# Multi-label Contrastive Predictive Coding

**Jiaming Song**
Stanford University
tsong@cs.stanford.edu

**Stefano Ermon**
Stanford University
ermon@cs.stanford.edu

## Abstract

Variational mutual information (MI) estimators are widely used in unsupervised representation learning methods such as contrastive predictive coding (CPC). A lower bound on MI can be obtained from a multi-class classification problem, where a critic attempts to distinguish a positive sample drawn from the underlying joint distribution from $(m-1)$ negative samples drawn from a suitable proposal distribution. Using this approach, MI estimates are bounded above by $\log m$, and could thus severely underestimate unless $m$ is very large. To overcome this limitation, we introduce a novel estimator based on a multi-label classification problem, where the critic needs to jointly identify *multiple* positive samples at the same time. We show that using the same amount of negative samples, multi-label CPC is able to exceed the $\log m$ bound, while still being a valid lower bound of mutual information. We demonstrate that the proposed approach is able to lead to better mutual information estimation, gain empirical improvements in unsupervised representation learning, and beat a current state-of-the-art knowledge distillation method over 10 out of 13 tasks.

## 1 Introduction

Learning efficient representations from data with minimal supervision is a critical problem in machine learning with significant practical impact [37, 12, 38, 41, 6]. Representations obtained using large amounts of unlabeled data can boost performance on downstream tasks where labeled data is scarce. This paradigm is already successful in a variety of domains; for example, representations trained on large amounts of unlabeled images can be used to improve performance on detection [55, 18, 9].

In the context of learning visual representations, contrastive objectives based on variational mutual information (MI) estimation are among the most successful ones [49, 3, 13, 40, 47]. One such approach, named Contrastive Predictive Coding (CPC, [49]), obtains a lower bound to MI via a multi-class classification problem. In CPC, a critic is generally trained to distinguish a pair of representations from two augmentations of the same image (positive), apart from $(m-1)$ pairs of representations from different images (negative). The representation network is then trained to increase the MI estimates given by the critic. This brings together the two representations from the positive pair and pushes apart the two representations from the negative pairs.

It has been empirically observed that factors leading to better MI estimates, such as training for more iterations and increasing the complexity of the critic [9, 10], can generally result in improvements over downstream tasks. In the context of CPC, increasing the number of negative samples per positive sample (i.e. increasing $m$) also helps with downstream performance [53, 18, 9, 47]. This can be explained from a mutual information estimation perspective that CPC estimates are upper bounded by $\log m$, so increasing $m$ could reduce bias when the actual mutual information is much higher [35]. However, due to constraints over compute, memory and data, there is a limit to how many negative samples we can obtain per positive sample.

In this paper, we propose generalizations to CPC that can increase the $\log m$ bound without additional computational costs, thus decreasing bias. We first generalize CPC through by re-weighting the influence of positive and negative samples in the underlying the classification problem. This increases the $\log m$ bound and leads to bias reduction, yet the re-weighted CPC objective is no longer guaranteed to be a lower bound to mutual information.

To this end, we introduce multi-label CPC (ML-CPC) which poses mutual information estimation as a multi-label classification problem. Instead of identifying one positive sample for each classification task (as in CPC), the critic now simultaneously identifies multiple positive samples that come from the same batch. We prove for ML-CPC that under certain choices of the weights, we can increase the $\log m$ bound and reduce bias, while guaranteeing that the new objective is still lower bounded by mutual information. This provides an practical algorithm whose upper bound is close to the theoretical upper limit by any distribution-free, high-confidence lower bound estimators of mutual information [36].

Re-weighted ML-CPC encompasses a range of mutual information lower bound estimators with different bias-variance trade-offs, which can be chosen with minimal impact on the computational costs. We demonstrate the effectiveness of re-weighted ML-CPC over CPC empirically on several tasks, including mutual information estimation, knowledge distillation and unsupervised representation learning. In particular, ML-CPC is able to beat the current state-of-the-art method in knowledge distillation [47] on 10 out of 13 distillation tasks for CIFAR-100.

## 2 Contrastive Predictive Coding and Mutual Information

In representation learning, we are interested in learning a (possibly stochastic) network $h : \mathcal{X} \to \mathcal{Y}$ that maps some data $\boldsymbol{x} \in \mathcal{X}$ to a compact representation $h(\boldsymbol{x}) \in \mathcal{Y}$. For ease of notation, we denote $p(\boldsymbol{x})$ as the data distribution, $p(\boldsymbol{x}, \boldsymbol{y})$ as the joint distribution for data and representations (denoted as $\boldsymbol{y}$), $p(\boldsymbol{y})$ as the marginal distribution of the representations, and $X, Y$ as the random variables associated with data and representations. The InfoMax principle [32, 4, 13] for learning representations considers variational maximization of the mutual information $I(X; Y)$:

$$I(X; Y) := \mathbb{E}_{(\boldsymbol{x}, \boldsymbol{y}) \sim p(\boldsymbol{x}, \boldsymbol{y})} \left[ \log \frac{p(\boldsymbol{x}, \boldsymbol{y})}{p(\boldsymbol{x}) p(\boldsymbol{y})} \right] \tag{1}$$

A variety of mutual information estimators with different bias-variance trade-offs have been proposed for representation learning [39, 48, 3, 40]. Contrastive predictive coding (CPC, also known as InfoNCE [49]), poses the MI estimation problem as an $m$-class classification problem. Here, the goal is to distinguish a *positive* pair $(\boldsymbol{x}, \boldsymbol{y}) \sim p(\boldsymbol{x}, \boldsymbol{y})$ from $(m-1)$ *negative* pairs $(\boldsymbol{x}, \overline{\boldsymbol{y}}) \sim p(\boldsymbol{x}) p(\boldsymbol{y})$. If the optimal classifier is able to distinguish positive and negative pairs easily, it means $\boldsymbol{x}$ and $\boldsymbol{y}$ are tied to each other, indicating high mutual information.

For a batch of $n$ positive pairs $\{(\boldsymbol{x}_i, \boldsymbol{y}_i)\}_{i=1}^{n}$, the CPC objective is defined as[1]:

$$L(g) := \mathbb{E} \left[ \frac{1}{n} \sum_{i=1}^{n} \log \frac{m \cdot g(\boldsymbol{x}_i, \boldsymbol{y}_i)}{g(\boldsymbol{x}_i, \boldsymbol{y}_i) + \sum_{j=1}^{m-1} g(\boldsymbol{x}_i, \overline{\boldsymbol{y}_{i,j}})} \right] \tag{2}$$

for some positive critic function $g : \mathcal{X} \times \mathcal{Y} \to \mathbb{R}_+$, where the expectation is taken over $n$ positive pairs $(\boldsymbol{x}_i, \boldsymbol{y}_i) \sim p(\boldsymbol{x}, \boldsymbol{y})$ and $n(m-1)$ negative pairs $(\boldsymbol{x}_i, \overline{\boldsymbol{y}_{i,j}}) \sim p(\boldsymbol{x}) p(\boldsymbol{y})$.

### 2.1 CPC is a lower bound to mutual information

Oord et al. [49] interpreted the CPC objective as a lower bound to MI, but only proved the case for a lower bound approximation of CPC, where a term containing $-\mathbb{E}[\log g]$ is replaced by $-\log \mathbb{E}[g]$; so their arguments alone cannot prove that CPC is a lower bound of mutual information. Poole et al. [40] proved a lower bound argument for the objective where $m = n$ and negative samples are tied to other positive samples in the same batch. To bridge the gap between theory (that CPC instantiates InfoMax) and practice (where negative samples can be chosen independently from positive samples of the same batch, such as MoCo [18]), we present another proof for the general CPC objective as

presented in $L(g)$. First, we show the following result for variational lower bounds of KL divergences between general distributions where batches of negative samples are used to estimate the divergence. Then, as mutual information is a KL divergence between two specific distributions, the lower bound argument for CPC simply follows.

**Theorem 1.** *For all probability measures $P, Q$ over sample space $\mathcal{X}$ such that $P \ll Q$, the following holds for all functions $r : \mathcal{X} \to \mathbb{R}_+$ and integers $m \geq 2$:*

$$D_{\mathrm{KL}}(P\|Q) \geq \mathbb{E}_{\boldsymbol{x} \sim P, \boldsymbol{y}_{1:m-1} \sim Q^{m-1}} \left[ \log \frac{m \cdot r(\boldsymbol{x})}{r(\boldsymbol{x}) + \sum_{i=1}^{m-1} r(\boldsymbol{y}_i)} \right]. \tag{3}$$

*Proof.* In Appendix A, using the variational representations of $f$-divergences [39] and Prop. 1. □

The argument about CPC being a lower bound to MI is simply a corollary of the above statement where $P$ is $p(\boldsymbol{x}, \boldsymbol{y})$ (joint) and $Q$ is $p(\boldsymbol{x})p(\boldsymbol{y})$ (product of marginals); we state the claim below.

**Corollary 1.** $\forall n \geq 1, m \geq 2$, $\forall g : \mathcal{X} \times \mathcal{Y} \to \mathbb{R}_+$, *the following is true: $L(g) \leq I(X;Y)$.*

Therefore, one can train $g$ and $h$ to maximize $L(g)$ (recall that $L$ depends on $h$ via $\boldsymbol{y} = h(\boldsymbol{x})$), which is guaranteed to be lower than $I(X;Y)$ in expectation.

## 2.2 CPC is an estimator with high bias

For finite $m$, since $g(\boldsymbol{x}_i, \boldsymbol{y}_i)$ appears in both the numerator and denominator of Equation (2) and $g$ is positive, the density ratio estimates can be no larger than $m$, and the value of $L(g)$ is thus upper bounded by $\log m$ [49]. While this is acceptable for certain low dimensional scenarios, this can lead to high-bias if the true mutual information is much larger than $\log m$. In fact, the required $m$ can be unacceptable in high dimensions since MI can scale linearly with dimension, which means an exponential number of negative samples are needed to achieve low bias.

For example, if $X$ and $Y$ are 1000-dimensional random variables where the marginal distribution for each dimension is standard Gaussian, and for each dimension $d$, $X_d$ and $Y_d$ has a correlation of 0.2, then the mutual information $I(X;Y)$ is around 20.5, which means that $m$ has to be greater than $4 \times 10^8$ in order for CPC estimates to approach this value. In comparison, state-of-the-art image representation learning methods use a $m$ that is around 65536 and representation dimensions between 128 to 2048 [53, 18, 9] due to batch size and memory limitations, as one would need a sizeable batch of positive samples in order to apply batch normalization [24].

## 2.3 Re-weighted Contrastive Predictive Coding

Under the computational limitations imposed by $m$ (i.e., we cannot obtain too many negative samples per positive sample), we wish to develop generalizations to CPC that reduce bias while still being lower bounds to the mutual information. We do not consider other types of estimators such as MINE [3] or NWJ [39] because they would exhibit high variance on the order of $O(e^{I(X;Y)})$ [44], and thus are much less stable to optimize.

One possible approach is to decrease the weights of the positive sample when calculating the sum in the denominator; this leads to the following objective, called $\alpha$-CPC:

$$L_\alpha(g) := \mathbb{E}\left[ \frac{1}{n} \sum_{i=1}^{n} \log \frac{m \cdot g(\boldsymbol{x}_i, \boldsymbol{y}_i)}{\alpha g(\boldsymbol{x}_i, \boldsymbol{y}_i) + \frac{m-\alpha}{m-1} \sum_{j=1}^{m-1} g(\boldsymbol{x}_i, \overline{\boldsymbol{y}_{i,j}})} \right] \tag{4}$$

where the positive sample is weighted by $\alpha$ and negative samples are weighted by $\frac{m-\alpha}{m-1}$. The purpose of adding weights to negative samples is to make sure the the weights sum to $m$, like in the original case where each sample has weight 1 and there are $m$ samples in total. Clearly, the original CPC objective is a special case when $\alpha = 1$.

On the one hand, $L_\alpha(g)$ is now upper bounded by $\log \frac{m}{\alpha}$, which is larger than $\log m$ when $\alpha \in (0, 1)$. Thus, $\alpha$-CPC has the potential to reduce bias when $\log m$ is much smaller than $I(X;Y)$. On the other hand, when we set a smaller $\alpha$, the variance of the estimator becomes larger, and the objective $L_\alpha(g)$ becomes more difficult to optimize [21, 22]. Therefore, selecting an appropriate $\alpha$ to balance the bias-variance trade-off is helpful for optimization of the objective in practice.

However, it is now possible for $L_\alpha(g)$ to be larger than $I(X;Y)$ as the number of classes $m$ grows to infinity, so optimizing $L_\alpha(g)$ does not necessarily recover a lower bound to mutual information. We illustrate this via the following example (more details in Appendix C).

**Example 1.** *Let $X, Y$ be two binary r.v.s such that $\Pr(X = 1, Y = 1) = \Pr(X = 0, Y = 0) = 0.5$. Then $I(X;Y) = \log 2 \approx 0.69$. However, when $\alpha = 0.5$ and $n = m = 3$, we can analytically compute $L_\alpha(g) \approx 0.72 \geq I(X;Y)$ for $g(x, y) = 1$ if $x = y$ and near $0$ otherwise.*

## 3 Multi-label Contrastive Predictive Coding

While $\alpha$-CPC could be useful empirically, we lack a principled way to select proper values of $\alpha$ as $L_\alpha(g)$ may no longer be a lower bound to mutual information. In the following sections, we propose an approach that allows us to achieve both, *i.e.*, for all $\alpha$ in a certain range (that only depends on $n$ and $m$), we can achieve an upper bound of $\log \frac{m}{\alpha}$ while ensuring that the objective is still a lower bound on mutual information. This allows us to select different values of $\alpha$ to reflect different preferences over bias and variance, all while keeping the computational cost identical.

We consider solving a "$nm$-class, $n$-label" classification problem, where given $n$ positive samples and $n(m-1)$ negative samples $\overline{\boldsymbol{y}_{j,k}} \sim p(\boldsymbol{y})$, we wish to jointly identify the top-$n$ samples that are most likely to be the positive ones. Concretely, this has the following objective function:

$$J(g) := \mathbb{E}\left[\frac{1}{n}\sum_{i=1}^{n}\log\frac{nm \cdot g(\boldsymbol{x}_i, \boldsymbol{y}_i)}{\sum_{j=1}^{n}g(\boldsymbol{x}_j, \boldsymbol{y}_j) + \sum_{j=1}^{n}\sum_{k=1}^{m-1}g(\boldsymbol{x}_j, \overline{\boldsymbol{y}_{j,k}})}\right] \qquad (5)$$

where the expectation is taken over the $n$ positive samples $(\boldsymbol{x}_i, \boldsymbol{y}_i) \sim p(\boldsymbol{x}, \boldsymbol{y})$ for $i \in [n]$ and the $n(m-1)$ negative samples $\overline{\boldsymbol{y}_{j,k}} \sim p(\boldsymbol{y})$ for $j \in [n], k \in [m-1]$. We call this *multi-label contrastive predictive coding* (ML-CPC), since the classifier now needs to predict $n$ positive labels from $nm$ options at the same time, instead of $1$ positive label from $m$ options as in traditional CPC (performed for $n$ times for a batch size of $n$).

**Distinctions from CPC** Despite its similarity compared to CPC (both are based on classification), we note that the multi-label perspective is fundamentally different from the CPC paradigm in three aspects, and cannot be treated as simply increasing the number of negative samples.

1. The ML-CPC objective value depends on the batch size $n$, whereas the CPC objective does not.
2. In CPC the positive pair and negative pairs share a same element ($\boldsymbol{x}_i$ in Eq.(2) where the positive sample is $(\boldsymbol{x}_i, \boldsymbol{y}_i)$), whereas in ML-CPC the negative pairs no longer have such restrictions; this could be useful for smaller datasets $\mathcal{D}$ when the number of possible negative pairs increases from $O(|\mathcal{D}|)$ to $O(|\mathcal{D}|^2)$.
3. The optimal critic for CPC is $g^\star = c(\boldsymbol{x}) \cdot p(\boldsymbol{x}, \boldsymbol{y})/(p(\boldsymbol{x})p(\boldsymbol{y}))$, where $c$ is any positive function of $\boldsymbol{x}$ [35]. In ML-CPC, different $\boldsymbol{x}$ values are tied within the same batch, so the optimal critic for ML-CPC is $g^\star = c \cdot p(\boldsymbol{x}, \boldsymbol{y})/(p(\boldsymbol{x})p(\boldsymbol{y}))$, where $c$ is a positive constant. As a result, ML-CPC reduces the amount of optimal solutions, and forces the similarity of *any* positive pair to be higher than that of *any* negative pair, unlike CPC where the positive pair only needs to have higher similarity than any negative pairs with the same $\boldsymbol{x}$.

**Computational cost of ML-CPC** To compute CPC with a batch size of $n$, one would need $nm$ critic evaluations and compute $n$ sums in the denominator, each over a different set of $m$ evaluations. To compute ML-CPC, one needs $nm$ critic evaluations, and compute $1$ sum over all $nm$ evaluations. Therefore, ML-CPC has almost the same computational cost compared to CPC which is $O(mn)$. We perform a similar analysis in Appendix A to show that evaluating the gradients of the objectives also has similar costs, so ML-CPC is computationally as efficient as CPC.

### 3.1 Re-weighted Multi-label Contrastive Predictive Coding

Similar to $\alpha$-CPC, we can modify the multi-label objective $J(g)$ by re-weighting the critic predictions, which results in the following objective called $\alpha$-ML-CPC:

$$J_\alpha(g) := \mathbb{E}\left[\frac{1}{n}\sum_{i=1}^{n}\log\frac{nm \cdot g(\boldsymbol{x}_i, \boldsymbol{y}_i)}{\alpha\sum_{j=1}^{n}g(\boldsymbol{x}_j, \boldsymbol{y}_j) + \frac{m-\alpha}{m-1}\sum_{j=1}^{n}\sum_{k=1}^{m-1}g(\boldsymbol{x}_j, \overline{\boldsymbol{y}_{j,k}})}\right] \qquad (6)$$

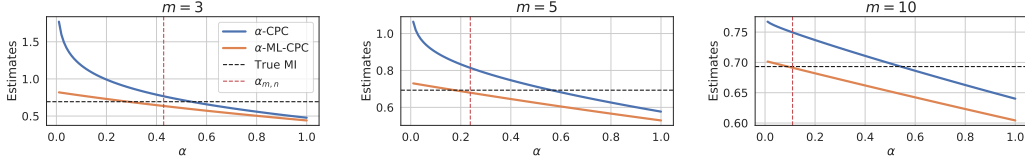

Figure 1: MI estimates with CPC and ML-CPC under different $\alpha$.

For $\alpha \in (0, 1)$, we down-weight the positive critic outputs by $\alpha$ and up-weight the negative critic outputs by $\frac{m-\alpha}{m-1}$ (similar to $\alpha$-CPC). Setting a smaller $\alpha$ has the potential to reduce bias, since the upper bound of $\log m$ is changed to $\log \frac{m}{\alpha}$, which is larger when $\alpha \in (0, 1)$. In contrast to $\alpha$-CPC, $J_\alpha(g)$ is now guaranteed to be a lower bound of mutual information for a wide range of $\alpha$, as we show in the following statements. Similar to the case of CPC, we first show a more general argument, for which the weighted ML-CPC is a special case.

**Theorem 2.** *For all probability measures $P, Q$ over sample space $\mathcal{X}$ such that $P \ll Q$, the following holds for all functions $r : \mathcal{X} \to \mathbb{R}_+$, integers $n \geq 1, m \geq 2$, and real numbers $\alpha \in [\frac{m}{n(m-1)+1}, 1]$:*

$$D_{\mathrm{KL}}(P\|Q) \geq \mathbb{E}_{\boldsymbol{x}_{1:n}\sim P^n, \boldsymbol{y}_{i,1:m-1}\sim Q^{m-1}} \left[ \frac{1}{n}\sum_{i=1}^{n} \log \frac{mn \cdot r(\boldsymbol{x}_i)}{\alpha \sum_{j=1}^{n} r(\boldsymbol{x}_j) + \frac{m-\alpha}{m-1}\sum_{k=1}^{m-1} r(\boldsymbol{y}_{j,k})} \right]. \quad (7)$$

*Proof.* In Appendix A, using the variational representations of $f$-divergences [39] and Prop. 2. $\quad\square$

The above theorem extends existing variational lower bound estimators of KL divergences (that are generally interpreted as binary classification [46, 39]) into a family of lower bounds that can be interpreted as multi-label classification. The argument about re-weighted ML-CPC being a lower bound to MI is simply a corollary where $P$ is $p(\boldsymbol{x}, \boldsymbol{y})$ and $Q$ is $p(\boldsymbol{x})p(\boldsymbol{y})$; we state the claim below.

**Corollary 2.** $\forall n \geq 1, m \geq 2$, *define* $\alpha_{m,n} = \frac{m}{n(m-1)+1}$. *If* $\alpha \in [\alpha_{m,n}, 1]$, *then* $\forall g : \mathcal{X} \times \mathcal{Y} \to \mathbb{R}_+$,

$$J_\alpha(g) \leq I(X; Y) \quad (8)$$

The above theorem shows that for an appropriate range of $\alpha$ values, the objective $J_\alpha(g)$ is still guaranteed to be a variational lower bound to mutual information, like the original CPC objective. Selecting $\alpha$ within this range results in estimators with different bias-variance trade-offs. Here, a smaller $\alpha$ could lead to low-bias high-variance estimates; this achieves a similar effect to increasing the number of classes $m$ to nearly $m/\alpha$, but without the actual additional computational costs that comes with obtaining more negative samples in CPC.

**Illustrative example**  We consider the case of $X, Y$ being binary and equal random variables in Example 1, where $I(X; Y) = \log 2 \approx 0.69$, the optimal critic $g$ is known, and both $L_\alpha(g)$ and $J_\alpha(g)$ can be computed in closed-form for any $\alpha$ and $g$ in $O(m)$ time (details in Appendix C). We plot the CPC (Eq.(4)) and ML-CPC (Eq.(6)) objectives with different choices of $\alpha$ and $m$ in Figure 1. The estimates of ML-CPC when $\alpha \geq \alpha_{m,n}$ are lower bounds to the ground truth MI, which indeed aligns with our theory.

Furthermore, in Figure 2 we illustrate the bias-variance trade-offs for CPC and $\alpha_{m,n}$-ML-CPC as we vary the number of classes $m$ (for simplicity, we choose $n = m$). Despite having slightly higher variance in the estimates, $\alpha_{m,n}$-ML-CPC has significantly less bias than CPC, which suggests that it is helpful in cases where lower bias is preferable than lower variance. In practice, the user could select different values of $\alpha$ to indicate the desired trade-off, all without having to change the number of negative samples and increase computational costs.

We include the pseudo-code and a PyTorch implementation to $\alpha$-ML-CPC in Appendix B.

## 4   Related Work

**Contrastive methods for representation learning**  The general principle of contrastive methods for representation learning encourages representations to be closer between "positive" pairs and

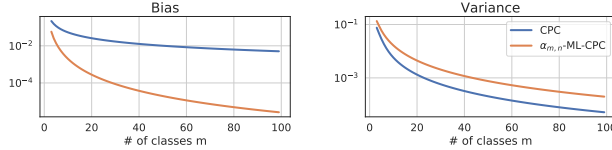

Figure 2: Bias-variance trade-offs for different $m$. Lower is better.

further between "negative" pairs, which has been applied to learning representations in various domains such as images [19, 53, 18, 9], words [37, 12], graphs [50] and videos [17]. Commonly used objectives include the logistic loss [37], margin triplet loss [42], the noise contrastive estimation loss [16] and other objectives based on variational lower bounds of mutual information, such as MINE [3] and CPC [49]. CPC-based approaches have gained much recent interest due to its superior performance in downstream tasks compared to other losses such as the logistic and margin loss [9].

**Variational mutual information estimators**  Estimating mutual information from samples is challenging [36, 54]. Most variational approaches to mutual information estimation are based on the Fenchel dual representation of $f$-divergences [39, 43], where a critic function is trained to learn the density ratio $p(\boldsymbol{x}, \boldsymbol{y})/(p(\boldsymbol{x})p(\boldsymbol{y}))$. These approaches mostly vary in terms of how the critics are modeled and optimized [2, 40], and exhibit different bias-variance trade-offs from these choices.

CPC would tend to underestimate the density ratio (since it is capped at $m$) and generally requires $O(e^{I(X;Y)})$ samples to achieve low bias; MINE [3] (based on the Donsker-Varadhan inequality [14]) is a biased estimator and requires $O(e^{I(X;Y)})$ samples to achieve low variance [44, 43]. Poole et al. [40] proposed an estimator that interpolates between two types of estimators, allowing for certain bias-variance trade-offs; this is relevant to our proposed re-weighted CPC in the sense that positive samples are down-weighted, but an additional baseline model is required during training. Through ML-CPC, we introduce a family of unbiased mutual information lower bound estimators, and reflect a wide range of bias-variance trade-offs without using more negative samples.

**Relevance to the limitations of mutual information lower bound estimators**  Furthermore, we note that ML-CPC is upper bounded by $\log(n(m-1)+1)$ for the smallest possible $\alpha$, which appears to be very close to (but smaller than) the general upper limit of $O(\log nm)$ that can be achieved by any distribution-free high-confidence lower bound on mutual information for $nm$ samples [36]. However, we note that the assumptions in [36] are slightly different to our settings, in the sense that they assumed complete access to the distribution $p(\boldsymbol{x}, \boldsymbol{y})$ and only required samples from $p(\boldsymbol{x})p(\boldsymbol{y})$, whereas we have to estimate $p(\boldsymbol{x}, \boldsymbol{y})$ from the samples as well; and the amount of samples we obtain from $p(\boldsymbol{x})p(\boldsymbol{y})$ is $n(m-1)$ instead of $nm$. We hypothesize that we can reach the theoretical limit with a method derived from ML-CPC, but leave it as an interesting future direction.

**Re-weighted softmax loss**  Generalizations to the softmax loss have been proposed in which different weights are assigned to different classes or samples [33, 34, 51], which are commonly used with regularization [7]. When the dataset has extremely imbalanced classes, higher weights are given to classes with less frequency [21, 22, 52] or classes with less effective samples [11]. Cao et al. [8] investigate re-weighting approaches that encourages large margins to the decision boundary for minority classes; such a context is also studied for detection [31] and segmentation [25] where class imbalance exists. Our work introduce re-weighting approaches to the context of unsupervised representation learning (where class labels do not exist in the traditional sense), where we aim for flexible bias-variance trade-offs in contrastive mutual information estimators.

## 5  Experiments

We evaluate our proposed methods on mutual information estimation, knowledge distillation and unsupervised representation learning. *To ensure fair comparisons are made, we only make adjustments to the training objective, and keep the remaining experimental setup identical to that of the baselines.* We describe details to the experimental setup in Appendix C. Our code is available at https://github.com/jiamings/ml-cpc.

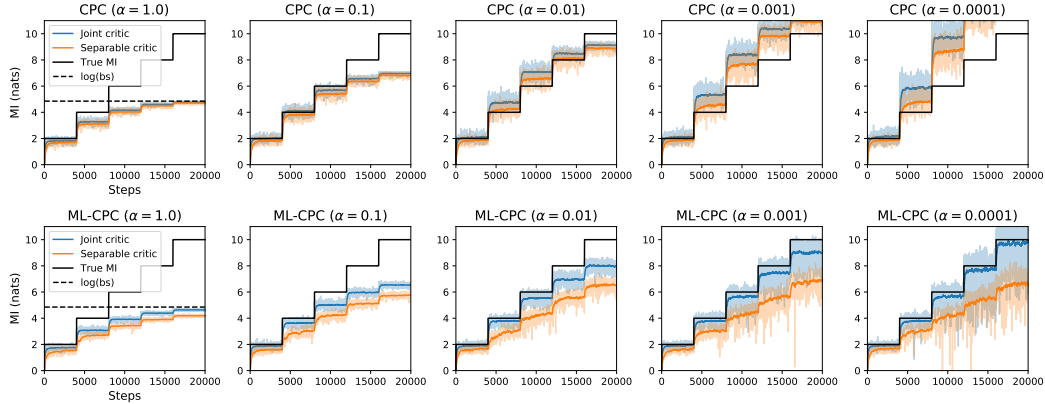

Figure 3: Mutual information estimation with CPC and ML-CPC with different choices of $\alpha$.

## 5.1 Mutual Information Estimation

**Setup** We first consider mutual information estimation between two correlated Gaussians of 20 dimensions, following the setup in [40, 44] where the ground truth mutual information is known and increases by 2 every 4k iterations, for a total of 20k iterations. We evaluate CPC and ML-CPC with different choices of $\alpha$ (ranging from 1.0 to 0.0001, which might not guarantee that they are lower bounds to mutual information) under two types of critic, named joint [3] and separable [49]. We use $m = n = 128$ in our experiments.

**Results** We illustrate the estimates and the ground truth MI in Figure 3. Both CPC and ML-CPC estimates are bounded by $\log m$ when $\alpha = 1$, which is no longer the case when we set smaller values of $\alpha$; however, as we decrease $\alpha$, CPC estimates are no longer guaranteed to be lower bounds to mutual information, whereas ML-CPC estimates still provide lower bound estimates in general. Moreover, a reduction in $\alpha$ for ML-CPC reduces bias at the cost of increasing variance, as the problem becomes more difficult with re-weighting. The time to compute 200 updates on a Nvidia 1080 Ti GPU with the a PyTorch implementation is $1.15 \pm 0.06$ seconds with CPC and $1.14 \pm 0.04$ seconds with ML-CPC, so the computational costs are indeed near identical.

## 5.2 Knowledge Distillation

**Setup** We apply re-weighted CPC and ML-CPC to knowledge distillation (KD, [20]), in which one neural network model (teacher) transfers its knowledge to another model (student, typically smaller) so that the student's performance is higher than training from labels alone. Contrastive representation distillation (CRD, [47]) is a state-of-the-art method that regularizes the student model so that its features have higher mutual information with that of the teacher; CRD is implemented via a type of noise contrastive estimation objective [16]. We replace this objective with CPC and ML-CPC, using different choices of $\alpha$ that are fixed throughout training, and *keeping the remaining hyperparameters identical to the CRD ones* in [47]. Two baselines are considered: the original KD objective in [20] and the state-of-the-art CRD objective in [47], since other baselines [29, 1, 23, 26] are shown to have inferior performance in general.

**Results** Following the procedure in [47], we evaluate over 13 different student-teacher pairs on CIFAR-100 [30]. The student and teacher have the same type of architecture in 7 cases and different types in 6 cases. We report top-1 test accuracy in Table 1 (same type) and Table 2 (different types), where each case is the mean evaluation from 3 random seeds. We omit the standard deviation across different random seeds of each setup to fit the table in the paper, but we note that deviation among different random seeds is fairly small (at around 0.05 to 0.1 for most cases). While CPC and ML-CPC are generally inferior to that of CRD when $\alpha = 1.0$ (this aligns with the observation in [47]), they outperform CRD in 10 out of 13 cases when a smaller $\alpha$ is selected.

To demonstrate the effect of improved performance of smaller $\alpha$, we evaluate average top-1 accuracies with $\alpha \in \{0.01, 0.05, 0.1, 0.2, 0.5, 1.0\}$ in Figure 4. Both CPC and ML-CPC are generally inferior

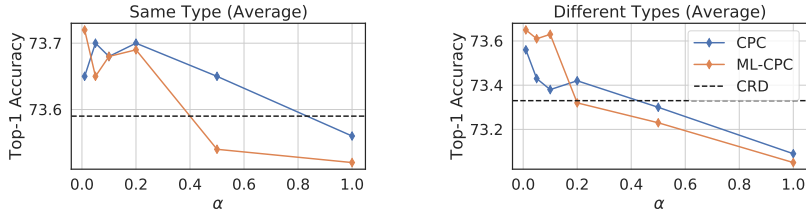

Figure 4: Ablation studies for KD with CPC and ML-CPC at different values of $\alpha$. Left: student and teacher are of the same type. Right: student and teacher are from different types.

to CRD when $\alpha = 1.0$ or $0.5$, but as we select smaller values of $\alpha$, they become superior to CRD and reaches the highest values at around $0.01$ to $0.05$, with ML-CPC being slightly better. Moreover, $n = 64, m = 16384$ so $\alpha_{m,n} \approx 0.015$, which achieves the lowest bias while ensuring ML-CPC to be a lower bound to MI. Thus this observation aligns with our claims on $\alpha_{m,n}$ in Theorem 2.

Table 1: Top-1 *test accuracy* (%) of students networks on CIFAR100 where the student and teacher networks are of the same type. ($\uparrow$) and ($\downarrow$) denotes superior and inferior performance relative to CRD. Each result is the mean of 3 random runs. $L_\alpha$ and $J_\alpha$ denote $\alpha$-CPC and $\alpha$-ML-CPC.

| Teacher | WRN-40-2 | WRN-40-2 | resnet56 | resnet110 | resnet110 | resnet32x4 | vgg13 |
| Student | WRN-16-2 | WRN-40-1 | resnet20 | resnet20 | resnet32 | resnet8x4 | vgg8 |
| --- | --- | --- | --- | --- | --- | --- | --- |
| Teacher | 75.61 | 75.61 | 72.34 | 74.31 | 74.31 | 79.42 | 74.64 |
| Student | 73.26 | 71.98 | 69.06 | 69.06 | 71.14 | 72.50 | 70.36 |
| KD | 74.92 | 73.54 | 70.66 | 70.67 | 73.08 | 73.33 | 72.98 |
| CRD | 75.48 | 74.14 | 71.16 | **71.46** | 73.48 | **75.51** | 73.94 |
| $L_{1.0}$ | 75.42 ($\downarrow$) | 74.16 ($\uparrow$) | 71.32 ($\uparrow$) | 71.39 ($\downarrow$) | 73.57 ($\uparrow$) | 75.50 ($\downarrow$) | 73.60 ($\downarrow$) |
| $L_{0.1}$ | 75.69 ($\uparrow$) | 74.17 ($\uparrow$) | 71.48 ($\uparrow$) | 71.38 ($\downarrow$) | 73.66 ($\uparrow$) | 75.41 ($\downarrow$) | 73.61 ($\downarrow$) |
| $J_{1.0}$ | 75.39 ($\downarrow$) | 74.18 ($\uparrow$) | 71.28 ($\uparrow$) | 71.28 ($\downarrow$) | 73.58 ($\uparrow$) | 75.32 ($\downarrow$) | 73.67 ($\downarrow$) |
| $J_{0.05}$ | 75.64 ($\uparrow$) | **74.27** ($\uparrow$) | 71.33 ($\uparrow$) | 71.24 ($\downarrow$) | 73.57 ($\uparrow$) | 75.50 ($\downarrow$) | **74.01** ($\uparrow$) |
| $J_{0.01}$ | **75.83** ($\uparrow$) | 74.24 ($\uparrow$) | **71.50** ($\uparrow$) | 71.27 ($\downarrow$) | **73.90** ($\uparrow$) | 75.37 ($\downarrow$) | 73.95 ($\uparrow$) |

Table 2: Top-1 *test accuracy* (%) of students networks on CIFAR100 where the student and teacher networks are from different types. ($\uparrow$) and ($\downarrow$) denotes superior and inferior performance relative to CRD. Each result is the mean of 3 random runs. $L_\alpha$ and $J_\alpha$ denote $\alpha$-CPC and $\alpha$-ML-CPC.

| Teacher | vgg13 | ResNet50 | ResNet50 | resnet32x4 | resnet32x4 | WRN-40-2 |
| Student | MobileNetV2 | MobileNetV2 | vgg8 | ShuffleNetV1 | ShuffleNetV2 | ShuffleNetV1 |
| --- | --- | --- | --- | --- | --- | --- |
| Teacher | 74.64 | 79.34 | 79.34 | 79.42 | 79.42 | 75.61 |
| Student | 64.60 | 64.60 | 70.36 | 70.50 | 71.82 | 70.50 |
| KD | 67.37 | 67.35 | 73.81 | 74.07 | 74.45 | 74.83 |
| CRD | **69.73** | 69.11 | 74.30 | 75.11 | 75.65 | 76.05 |
| $L_{1.0}$ | 69.24 ($\downarrow$) | 69.02 ($\downarrow$) | 73.66 ($\downarrow$) | 75.00 ($\downarrow$) | 75.93 ($\uparrow$) | 75.72 ($\downarrow$) |
| $L_{0.1}$ | 69.26 ($\downarrow$) | 69.33 ($\uparrow$) | 74.24 ($\downarrow$) | 75.34 ($\uparrow$) | 76.01 ($\uparrow$) | 76.12 ($\uparrow$) |
| $J_{1.0}$ | 68.92 ($\downarrow$) | 68.80 ($\downarrow$) | 73.65 ($\downarrow$) | 75.39 ($\uparrow$) | 75.88 ($\uparrow$) | 75.70 ($\downarrow$) |
| $J_{0.05}$ | 69.25 ($\downarrow$) | **70.04** ($\uparrow$) | **74.84** ($\uparrow$) | 75.51 ($\uparrow$) | 76.24 ($\uparrow$) | 76.03 ($\uparrow$) |
| $J_{0.01}$ | 69.25 ($\downarrow$) | 69.90 ($\uparrow$) | 74.81 ($\uparrow$) | **75.47** ($\uparrow$) | **76.04** ($\uparrow$) | **76.19** ($\uparrow$) |

## 5.3 Representation Learning

**Setup** Finally, we consider ML-CPC for unsupervised representation learning as a replacement to CPC. We follow the experiment procedures in MoCo-v2 [10] (which used the CPC objective), where negative samples are obtained from a key encoder that updates more slowly than the representation

network. We use the "linear evaluation protocol" where the learned representations are evaluated via the test top-1 accuracy when a linear classifier is trained to predict labels from representations. Different from knowledge distillation, we do not have labels and fixed teacher representations, so the problem becomes much more difficult and using small values of $\alpha$ alone will lead to high variance in initial estimates which could hinder the final performance. To this end, we use a curriculum learning [5] approach where we select $\alpha$ values from high to low throughout training: higher $\alpha$ has higher bias, lower variance and easier to learn, whereas lower $\alpha$ has lower bias, higher variance and harder to learn. For ML-CPC, we consider 4 types of geometrically decreasing schedules for $\alpha$: fixed at 1.0; from 2.0 to 0.5; from 5.0 to 2.0; and from 10.0 to 0.1; so $\alpha = 1.0$ for all cases when we reached half of the training epochs. We use the same values for other hyperparameters as those used in the MoCo-v2 CPC baseline (more details in Appendix C).

**Results** We show the top-1 accuracy of the learned representations under the linear evaluation protocol in Table 3 for CIFAR10 and CIFAR100. While the original ML-CPC objective (denoted as $J_{1.0} \rightarrow J_{1.0}$) already outperforms the CPC baseline in most cases, we observe that using a curriculum from easy to hard objective has the potential to further improve performance of the representations. Notably, the $J_{10.0} \rightarrow J_{0.1}$ schedule improves the performance on both datasets by almost 2.5 percent when trained for 200 epochs, which demonstrates its effectiveness when the number of epochs used during training is limited.

Table 3: Top-1 accuracy of unsupervised representation learning under the linear evaluation protocol.

(a) CIFAR-10

| Epochs | 200 | 500 | 1000 |
|---|---|---|---|
| $L_{1.0}$ | 83.28 | 89.31 | 91.20 |
| $J_{1.0} \rightarrow J_{1.0}$ | 83.61 (↑) | 89.43 (↑) | 91.48 (↑) |
| $J_{2.0} \rightarrow J_{0.5}$ | 84.31 (↑) | 89.47 (↑) | 91.43 (↑) |
| $J_{5.0} \rightarrow J_{0.2}$ | 85.52 (↑) | **89.85** (↑) | 91.50 (↑) |
| $J_{10.0} \rightarrow J_{0.1}$ | **86.16** (↑) | 89.49 (↑) | **91.86** (↑) |

(b) CIFAR-100

| Epochs | 200 | 500 | 1000 |
|---|---|---|---|
| $L_{1.0}$ | 61.42 | 67.72 | 69.63 |
| $J_{1.0} \rightarrow J_{1.0}$ | 61.80 (↑) | 67.68 (↓) | **70.85** (↑) |
| $J_{2.0} \rightarrow J_{0.5}$ | 62.92 (↑) | 68.01 (↑) | 70.22 (↑) |
| $J_{5.0} \rightarrow J_{0.2}$ | 63.58 (↑) | **68.04** (↑) | 70.07 (↑) |
| $J_{10.0} \rightarrow J_{0.1}$ | **64.05** (↑) | 67.94 (↑) | 70.03 (↑) |

In Table 4, we include additional results for ImageNet under a compute-constrained scenario, where the representations are trained for only 30 epochs on a ResNet-18 architecture. Similar to the observations in CIFAR-10, we observe improvements in terms of linear classification accuracy of the learned representations. This demonstrates that the curriculum learning approach (specific to ML-CPC with re-weighting schedules, where the objective remains a lower bound to mutual information) could be useful to unsupervised representation learning in general.

## 6 Conclusion

In this paper, we proposed multi-label contrastive predictive coding for representation learning, which provides a generalization to contrastive predictive coding via multi-label classification. Re-weighted ML-CPC is able to enjoy less bias while being a lower bound to mutual information. Our upper bounds for the smallest $\alpha$ is close to the theoretical limit [36] of any distribution-free high-confidence lower bound on mutual information estimation. We demonstrate the effectiveness of ML-CPC on mutual information, knowledge distillation and unsupervised representation learning.

It would be interesting to further apply this method to other application domains, investigate alternative methods to control the re-weighting procedure (such as using angular margins [33]), and develop more principled approaches towards curriculum learning for unsupervised representation learning. From a theoretical standpoint, it is also interesting to formally investigate the bias-variance trade-off of ML-CPC, and see whether simple modifications to ML-CPC based on a slightly different assumption over $p(\boldsymbol{x}, \boldsymbol{y})$ could approach the theoretical limit by McAllester and Stratos [36].

Table 4: ImageNet representation learning for 30 epochs.

| Objective | $L_{1.0}$ | $J_{1.0} \rightarrow J_{1.0}$ | $J_{2.0} \rightarrow J_{0.5}$ | $J_{5.0} \rightarrow J_{0.2}$ | $J_{10.0} \rightarrow J_{0.1}$ |
|---|---|---|---|---|---|
| Top1 | 43.45 | 43.24 (↓) | 43.52 (↑) | **43.86** (↑) | 43.81 (↑) |
| Top5 | 67.42 | 67.43 (↑) | **67.82** (↑) | 67.67 (↑) | 67.71 (↑) |

## Broader Impact

Unsupervised representation learning approaches have driven a lot of the recent progresses in many applications such as computer vision [18] and natural language processing [12]. However, training effective unsupervised learning models would require vast amounts of growing resources including data, compute and energy. For example, the recent GPT-3 [6] model with 175B parameters is trained on a dataset with 400B tokens and consumes thousands of Petaflops-s/days. As a result, it becomes ever increasingly difficult for those who does not have access to such resources to compete, leaving much progress in deep unsupervised representation learning at the hands of a few large organizations.

In order to further democratize AI, it has become crucial to develop efficient methods that can be reproduced by most individuals with low cost, from modeling, training to inference. Our work aims to make a very tiny step in this direction, where we have demonstrated improvements to existing algorithms under the same computational budget constraints. In particular, we are able to significantly improve the representation learning capability of a model under very limited computational budgets. Our method is also useful for other applications where estimating mutual information is involved, such as information bottleneck.

Nevertheless, our method is not agnostic to existing biases in the dataset, so there is a potential danger that any bias that are inherent in the data collection process are also exhibited in the learned representations, such as bias against minority groups [45]. Our method also does not consider the potential risks of adversarial examples [15], which could be designed to sabotage certain downstream tasks; as well as data poisoning [28], which could harm the quality of the learned representations. We encourage researchers to further think about these safety concerns of unsupervised representation learning, since unsupervised data sources are more susceptible to malevolent sources who exploit the shortage of regulators overlooking the data.

## Acknowledgements

The authors would like to thank David McAllester for suggesting the generalization to KL divergences between any two distributions, Shengjia Zhao for helpful discussions over the ideas, Alessandro Sordoni for identifying a typo in the proof, and the anonymous reviewers for their constructive feedback. This research was supported by NSF (#1651565, #1522054, #1733686), ONR (N00014-19-1-2145), AFOSR (FA9550-19-1-0024), Amazon AWS, and FLI.

## Footnotes

[1]We suppress the dependencies on $n$ and $m$ in $L(g)$ (and in subsequent objectives) for conciseness.

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
