[Supplementary Material]

# A Proofs

## A.1 Preliminary Lemma and Propositions

To prove the main results, we need the following Lemma and Propositions 1 and 2. The Lemma is a special case to the dual representation of $f$-divergences discussed in [39].

**Lemma 1** (Nguyen et al. [39]). $\forall P, Q \in \mathcal{P}(\mathcal{X})$ *such that* $P \ll Q$,

$$D_{\mathrm{KL}}(P\|Q) = \sup_{T \in L^\infty(Q)} \mathbb{E}_P[T] - \mathbb{E}_Q[e^T] + 1 \tag{9}$$

*Proof.* (Sketch) Please refer to [39] for a more formal proof.

Denote $f(t) = t \log t$ whose convex conjugate is $f^\star(u) = \exp(u-1)$, we have that

$$D_{\mathrm{KL}}(P\|Q) = \mathbb{E}_{\boldsymbol{x} \sim Q} \left[ f\left( \frac{\mathrm{d}P}{\mathrm{d}Q}(\boldsymbol{x}) \right) \right] = \mathbb{E}_{\boldsymbol{x} \sim Q} \left[ \sup_u u \cdot \frac{\mathrm{d}P}{\mathrm{d}Q}(\boldsymbol{x}) - f^\star(u) \right] \tag{10}$$

$$= \sup_{T \in L^\infty(Q)} \mathbb{E}_{\boldsymbol{x} \sim Q} \left[ T(\boldsymbol{x}) \cdot \frac{\mathrm{d}P}{\mathrm{d}Q}(\boldsymbol{x}) - f^\star(T(\boldsymbol{x})) \right] \tag{11}$$

$$= \sup_{T \in L^\infty(Q)} \mathbb{E}_{\boldsymbol{x} \sim P}[T(\boldsymbol{x})] - \mathbb{E}_{\boldsymbol{x} \sim Q}[f^\star(T(\boldsymbol{x}))] \tag{12}$$

$$= \sup_{T \in L^\infty(Q)} \mathbb{E}_{\boldsymbol{x} \sim P}[T(\boldsymbol{x}) + 1] - \mathbb{E}_{\boldsymbol{x} \sim Q}[\exp(T(\boldsymbol{x}))], \tag{13}$$

which completes the proof. $\square$

**Proposition 1.** *For all positive integers* $n \geq 1, m \geq 2$, *and for any collection of positive random variables* $\{X_i\}_{i=1}^n$, $\{\overline{X_{i,j}}\}_{j=1}^m$ *such that* $\forall i \in [n]$, $X_i, \overline{X_{i,1}}, \overline{X_{i,2}}, \dots, \overline{X_{i,m-1}}$ *are exchangeable, then* $\forall \alpha \in (0, \frac{2m}{m+1}]$, *the following is true:*

$$\mathbb{E}\left[ \frac{1}{n} \sum_{i=1}^n \frac{mX_i}{\alpha X_i + \frac{m-\alpha}{m-1} \sum_{j=1}^{m-1} \overline{X_{i,j}}} \right] \leq \frac{1}{\alpha}. \tag{14}$$

*Proof.* First, for $\alpha \in (0, 2m/(m+1)]$ we have:

$$n\mathbb{E}\left[ \frac{1}{n} \sum_{i=1}^n \frac{mX_i}{\alpha X_i + \frac{m-\alpha}{m-1} \sum_{j=1}^{m-1} \overline{X_{i,j}}} \right] \tag{15}$$

$$= \mathbb{E}\left[ \sum_{i=1}^n \frac{m\frac{m-1}{m-\alpha}X_i}{(\Sigma_i) - \left(1 - \frac{m-1}{m-\alpha}\alpha\right)X_i} \right] \tag{16}$$

$$= \mathbb{E}\left[ \sum_{i=1}^n \frac{m\frac{m-1}{m-\alpha}}{\Sigma_i/X_i - (1 - \frac{m-1}{m-\alpha}\alpha)} \right] \tag{17}$$

$$= m\frac{m-1}{m-\alpha}\mathbb{E}\left[ \sum_{i=1}^n \sum_{p=0}^\infty \left(\frac{X_i}{\Sigma_i}\right)^{p+1} \left(1 - \frac{m-1}{m-\alpha}\alpha\right)^p \right] \quad \text{(Taylor expansion)} \tag{18}$$

$$= m\frac{m-1}{m-\alpha} \sum_{i=1}^n \sum_{p=0}^\infty \mathbb{E}\left[ \left(\frac{X_i}{\Sigma_i}\right)^{p+1} \right] \left(1 - \frac{m-1}{m-\alpha}\alpha\right)^p \tag{19}$$

where we simplify the notation with $\Sigma_i := X_i + \sum_{j=1}^{m-1} \overline{X_{i,j}}$. Furthermore, we note that the Taylor series converges because $(1 - \frac{m-1}{m-\alpha}\alpha) \in (-1, 1)$.

Since the random variables are exchangeable, switching the ordering of $X_i, \overline{X_{i,1}}, \dots, \overline{X_{i,m-1}}$ does not affect the joint distribution, and the summing function is permutation invariant. Therefore, for all

$i \in [n], p \geq 0$,

$$\mathbb{E}\left[\left(\frac{X_i}{\Sigma_i}\right)^{p+1}\right] = \frac{1}{m}\mathbb{E}\left[\left(\frac{X_i}{\Sigma_i}\right)^{p+1} + \sum_{j=1}^{m-1}\left(\frac{\overline{X_{i,j}}}{\Sigma_i}\right)^{p+1}\right] \tag{20}$$

$$\leq \frac{1}{m}\mathbb{E}\left[\left(\frac{X_i}{\Sigma_i} + \sum_{j=1}^{m-1}\frac{\overline{X_{i,j}}}{\Sigma_i}\right)^{p+1}\right] = \frac{1}{m} \tag{21}$$

where the last inequality comes from the fact that $\left(X_i + \sum_{j=1}^{m-1}\overline{X_{i,j}}\right)/\Sigma_i = 1$ and all the random variables are positive. Continuing from Eq.(19), we have:

$$n\mathbb{E}\left[\frac{1}{n}\sum_{i=1}^{n}\frac{mX_i}{\alpha X_i + \frac{m-\alpha}{m-1}\sum_{j=1}^{m-1}\overline{X_{i,j}}}\right] \tag{22}$$

$$\leq nm\frac{m-1}{m-\alpha}\sum_{p=0}^{\infty}\frac{1}{m}\left(1 - \frac{m-1}{m-\alpha}\alpha\right)^p = \frac{m-1}{m-\alpha}\frac{n}{\alpha\frac{m-1}{m-\alpha}} = \frac{n}{\alpha} \tag{23}$$

Dividing both sides by $n$ completes the proof for $\alpha \in (0, \frac{2m}{m+1}]$. $\qquad\square$

**Proposition 2.** $\forall n \geq 1, m \geq 2$, and for any collection of positive random variables $\{X_i\}_{i=1}^n$, $\{\overline{X_{i,j}}\}_{j=1}^m$ such that $\forall i \in [n]$, $X_i, \overline{X_{i,1}}, \overline{X_{i,2}}, \dots, \overline{X_{i,m-1}}$ are exchangeable, then $\forall\alpha \in [1, \frac{m}{2}]$,

$$\mathbb{E}\left[\frac{1}{n}\sum_{i=1}^{n}\frac{mX_i}{\alpha X_i + \frac{m-\alpha}{m-1}\sum_{j=1}^{m-1}\overline{X_{i,j}}}\right] \leq 1 \tag{24}$$

*Proof.* The case for $\alpha \in [1, \frac{2m}{m+1}]$ is apparent from Proposition 1.

For $\alpha \in (2m/(m+1), m/2]$, we have for all $t \in [m-1]$:

$$\mathbb{E}\left[\frac{1}{n}\sum_{i=1}^{n}\frac{mX_i}{\alpha X_i + \frac{m-\alpha}{m-1}\sum_{j=1}^{m-1}\overline{X_{i,j}}}\right] \tag{25}$$

$$\leq \mathbb{E}\left[\frac{1}{n}\frac{1}{m-1}\sum_{i=1}^{n}\sum_{j=1}^{m-1}\frac{mX_i}{\alpha X_i + \sum_{k=j}^{j+t-1}\overline{X_{i,k}}}\right] \tag{26}$$

$$= \mathbb{E}\left[\frac{1}{n}\frac{1}{m-1}\sum_{i=1}^{n}\sum_{j=1}^{m-1}\frac{tX_i}{\frac{t\alpha}{m}X_i + \frac{t-\frac{t\alpha}{m}}{t-1}\sum_{k=j}^{j+t-1}\overline{X_{i,k}}}\right] \tag{27}$$

where we define $\overline{X_{i,k}} = \overline{X_{i,k-(m-1)}}$ when $k > (m-1)$ and use the concavity of the inverse function (or equivalently the HM-AM inequality) to establish Eq.(26). For any $\alpha \in (2m/(m+1), m/2]$, we can choose $t$ to be any integer from the interval $[\frac{m}{\alpha}, \frac{2m}{\alpha} - 1]$; we note that such an integer always exists because the length of the interval is greater or equal to 1:

$$\frac{2m}{\alpha} - 1 - \frac{m}{\alpha} = \frac{m}{\alpha} - 1 \geq 1$$

Then we can apply the result in Proposition 1, for $t$ samples and the new $\alpha$ being $\frac{t\alpha}{m}$; from our construction of $t$, this satisfies the condition in Proposition 1 that:

$$1 \leq \frac{t\alpha}{m} \leq \frac{2t}{t+1}$$

Therefore we can apply Proposition 1 to a valid choice of $t$ to obtain

$$\mathbb{E}\left[\frac{1}{n}\sum_{i=1}^{n}\frac{mX_i}{\alpha X_i + \frac{m-\alpha}{m-1}\sum_{j=1}^{m-1}\overline{X_{i,j}}}\right]$$

$$\leq \mathbb{E}\left[\frac{1}{n}\frac{1}{m-1}\sum_{i=1}^{n}\sum_{j=1}^{m-1}\frac{tX_i}{\frac{t\alpha}{m}X_i + \frac{t-\frac{t\alpha}{m}}{t-1}\sum_{k=j}^{j+t-1}\overline{X_{i,k}}}\right] \leq \frac{m}{t\alpha} \leq 1$$

which proves the result. $\qquad\square$

## A.2 Proof for CPC

**Theorem 1.** *For all probability measures $P, Q$ over sample space $\mathcal{X}$ such that $P \ll Q$, the following holds for all functions $r : \mathcal{X} \to \mathbb{R}_+$ and integers $m \geq 2$:*

$$D_{\mathrm{KL}}(P\|Q) \geq \mathbb{E}_{\boldsymbol{x} \sim P, \boldsymbol{y}_{1:m-1} \sim Q^{m-1}} \left[ \log \frac{m \cdot r(\boldsymbol{x})}{r(\boldsymbol{x}) + \sum_{i=1}^{m-1} r(\boldsymbol{y}_i)} \right]. \tag{3}$$

*Proof.* From Lemma 1, we have that:

$$D_{\mathrm{KL}}(P\|Q) \tag{28}$$

$$\geq \mathbb{E}_{\boldsymbol{y}_{1:m-1} \sim Q^{m-1}} \left[ \mathbb{E}_{\boldsymbol{x} \sim P} \left[ \log \frac{m \cdot r(\boldsymbol{x})}{r(\boldsymbol{x}) + \sum_{i=1}^{m-1} r(\boldsymbol{y}_i)} \right] - \mathbb{E}_{\boldsymbol{x} \sim Q} \left[ \frac{m \cdot r(\boldsymbol{x})}{r(\boldsymbol{x}) + \sum_{i=1}^{m-1} r(\boldsymbol{y}_i)} \right] + 1 \right]$$

$$\geq \mathbb{E}_{\boldsymbol{y}_{1:m-1} \sim Q^{m-1}} \left[ \mathbb{E}_{\boldsymbol{x} \sim P} \left[ \log \frac{m \cdot r(\boldsymbol{x})}{r(\boldsymbol{x}) + \sum_{i=1}^{m-1} r(\boldsymbol{y}_i)} \right] \right] - 1 + 1 \tag{29}$$

$$= \mathbb{E}_{\boldsymbol{x} \sim P, \boldsymbol{y}_{1:m-1} \sim Q^{m-1}} \left[ \log \frac{m \cdot r(\boldsymbol{x})}{r(\boldsymbol{x}) + \sum_{i=1}^{m-1} r(\boldsymbol{y}_i)} \right], \tag{30}$$

where the second inequality comes from Proposition 1 where $\boldsymbol{x} \sim Q$ and $\boldsymbol{y}_{1:m-1} \sim Q^{m-1}$ are exchangeable, thus proving the statement. $\qquad\square$

## A.3 Proof for ML-CPC

**Theorem 2.** *For all probability measures $P, Q$ over sample space $\mathcal{X}$ such that $P \ll Q$, the following holds for all functions $r : \mathcal{X} \to \mathbb{R}_+$, integers $n \geq 1, m \geq 2$, and real numbers $\alpha \in [\frac{m}{n(m-1)+1}, 1]$:*

$$D_{\mathrm{KL}}(P\|Q) \geq \mathbb{E}_{\boldsymbol{x}_{1:n} \sim P^n, \boldsymbol{y}_{i,1:m-1} \sim Q^{m-1}} \left[ \frac{1}{n} \sum_{i=1}^{n} \log \frac{mn \cdot r(\boldsymbol{x}_i)}{\alpha \sum_{j=1}^{n} r(\boldsymbol{x}_j) + \frac{m-\alpha}{m-1} \sum_{k=1}^{m-1} r(\boldsymbol{y}_{j,k})} \right]. \tag{7}$$

*Proof.* First, we have

$$\mathbb{E}\left[ \frac{1}{n} \sum_{i=1}^{n} \log \frac{nm \cdot r(\boldsymbol{x}_i)}{\alpha \sum_{j=1}^{n} r(\boldsymbol{x}_j) + \frac{m-\alpha}{m-1} \sum_{j=1}^{n} \sum_{k=1}^{m-1} r(\boldsymbol{y}_{j,k})} \right] \tag{31}$$

$$= \mathbb{E}\left[ \frac{1}{n} \sum_{i=1}^{n} \log(nm \cdot r(\boldsymbol{x}_i)) - \log\left( \alpha \sum_{j=1}^{n} r(\boldsymbol{x}_j) + \frac{m-\alpha}{m-1} \sum_{j=1}^{n} \sum_{k=1}^{m-1} r(\boldsymbol{y}_{j,k}) \right) \right]$$

$$\leq \mathbb{E}\left[ \frac{1}{n} \sum_{i=1}^{n} \log(nm \cdot r(\boldsymbol{x}_i)) - \log\left( n\alpha r(\boldsymbol{x}_i) + \frac{m-\alpha}{m-1} \sum_{j=1}^{n} \sum_{k=1}^{m-1} r(\boldsymbol{y}_{j,k}) \right) \right] \tag{32}$$

$$= \mathbb{E}\left[ \frac{1}{n} \sum_{i=1}^{n} \frac{nm \cdot r(\boldsymbol{x}_i)}{n\alpha r(\boldsymbol{x}_i) + \frac{m-\alpha}{m-1} \sum_{j=1}^{n} \sum_{k=1}^{m-1} r(\boldsymbol{y}_{j,k})} \right] \tag{33}$$

$$\leq D_{\mathrm{KL}}(P\|Q) - 1 + \mathbb{E}_{\boldsymbol{x}_{1:n} \sim Q^n} \left[ \frac{1}{n} \sum_{i=1}^{n} \frac{nm \cdot r(\boldsymbol{x}_i)}{n\alpha r(\boldsymbol{x}_i) + \frac{m-\alpha}{m-1} \sum_{j=1}^{n} \sum_{k=1}^{m-1} r(\boldsymbol{y}_{j,k})} \right] \tag{34}$$

where we use Jensen's inequality over log in Eq.(32) and Lemma 1 in Eq.(34).

Since $\boldsymbol{x}_i \sim Q$ and all the $\boldsymbol{y}_{j,k}$ are $(n(m-1)+1)$ exchangeable random variables, and

$$m \geq 2, \alpha \in \left[ \frac{m}{n(m-1)+1}, 1 \right] \quad \Rightarrow \quad \frac{n(m-1)+1}{m}\alpha \in \left[ 1, \frac{n(m-1)+1}{2} \right],$$

we can apply Proposition 2 to the $(n(m-1)+1)$ exchangeable variables

$$\mathbb{E}_{\boldsymbol{x}_{1:n} \sim Q^n} \left[ \frac{1}{n} \sum_{i=1}^{n} \frac{nm \cdot r(\boldsymbol{x}_i)}{n\alpha r(\boldsymbol{x}_i) + \frac{m-\alpha}{m-1} \sum_{j=1}^{n} \sum_{k=1}^{m-1} r(\boldsymbol{y}_{j,k})} \right]$$

$$= \mathbb{E}_{\boldsymbol{x}_{1:n} \sim Q^n} \left[ \frac{1}{n} \sum_{i=1}^{n} \frac{(n(m-1)+1) \cdot r(\boldsymbol{x}_i)}{\frac{n(m-1)+1}{m}\alpha r(\boldsymbol{x}_i) + \frac{m - \frac{n(m-1)+1}{nm}\alpha}{m-1} \sum_{j=1}^{n} \sum_{k=1}^{m-1} r(\boldsymbol{y}_{j,k})} \right] \le 1$$

Combining the above with Eq.(34), proves the result for the given range of $\alpha$. $\qquad\square$

### A.4   Time complexity of gradient calculation in ML-CPC

Suppose $g$ is a neural network parametrized by $\theta$, then the gradient to the ML-CPC objective is

$$\nabla_\theta J(g_\theta) = \mathbb{E} \left[ \frac{1}{n} \sum_{i=1}^{n} \frac{\nabla_\theta g_\theta(\boldsymbol{x}_i, \boldsymbol{y}_i)}{g_\theta(\boldsymbol{x}_i, \boldsymbol{y}_i)} - \frac{\sum_{j=1}^{n} \nabla_\theta g_\theta(\boldsymbol{x}_j, \boldsymbol{y}_j) + \sum_{j=1}^{n} \sum_{k=1}^{m-1} \nabla_\theta g_\theta(\boldsymbol{x}_j, \overline{\boldsymbol{y}_{j,k}})}{\sum_{j=1}^{n} g(\boldsymbol{x}_j, \boldsymbol{y}_j) + \sum_{j=1}^{n} \sum_{k=1}^{m-1} g(\boldsymbol{x}_j, \overline{\boldsymbol{y}_{j,k}})} \right]$$

Computing the gradient through the an empirical estimate of $J(g_\theta)$ requires us to perform back-propagation through all $nm$ critic evaluations, which is identical to the amount of back-propagation passes needed for CPC. So the time complexity to compute the ML-CPC gradient is $\mathcal{O}(nm)$.

## B   Pseudo-code and PyTorch implementation to ML-CPC

We include a PyTorch implementation to $\alpha$-ML-CPC as follows.

```
def ml_cpc(logits, alpha):
    """
    We assume that logits are of shape (n, m),
    and the predictions over positive are logits[:, 0].
    Alternatively, one can use kl_div() to ensure that the loss is non-negative.
    """
    n, m = logits.size(0), logits.size(1)
    beta = (m - alpha) / (m - 1)
    pos = logits.select(1, 0)
    neg = logits.narrow(1, 1, m)
    denom = torch.cat([pos + torch.log(torch.tensor(alpha)).float(),
                       neg + torch.log(torch.tensor(beta)).float()], dim=1)
    denom = denom.logsumexp(dim=1).logsumexp(dim=0)
    loss = denom - pos.sum()
    return loss / n
```

To ensure that the loss value is non-negative, one can alternatively use the `kl_div()` function that evaluates the KL divergence between the predicted label distribution with a ground truth label distribution. This is equivalent to the negative of the $\alpha$-ML-CPC objective shifted by a constant. We describe this idea in the following algorithm.

---
**Algorithm 1** Pseudo-code for $\alpha$-ML-CPC

---
1: **Input**: the critic $g$, input values $\boldsymbol{x}_i$, $\boldsymbol{y}_i$, $\overline{\boldsymbol{y}_{j,k}}$
2: **Output**: shifted negative objective value $J_\alpha(g)$ for optimization
3: Compute logit values $\ell_{i,i} = \log g(\boldsymbol{x}_i, \boldsymbol{y}_i) + \log \alpha$ and $\overline{\ell_{j,k}} = \log g(\boldsymbol{x}_j, \overline{\boldsymbol{y}_{j,k}}) + \log \frac{m-\alpha}{m-1}$.
4: Compute the normalization value $Z = \sum_i \exp(\ell_{i,i}) + \sum_{j,k} \exp(\overline{\ell_{j,k}})$.
5: Compute the predicted probabilities $p'_{i,i} = \exp(\ell_{i,i})/Z$, $\overline{p'_{j,k}} = \exp(\overline{\ell_{j,k}})/Z$
6: Assign the ground truth probabilities $p_{i,i} = 1/n$, $\overline{p_{j,k}} = 0$.
7: Compute the KL divergence between $p$ and $p'$.

---

# C Experimental Details

## C.1 Binary simulation experiments

Let $X, Y$ be two binary r.v.s such that $\Pr(X = 1, Y = 1) = p$, $\Pr(X = 0, Y = 0) = 1 - p$. We can simulate the case of a batch size of $n$ with $n - 1$ negative samples. For the example of CPC, we have:

$$L(g) := \mathbb{E}\left[ \frac{1}{n} \sum_{i=1}^{n} \log \frac{n \cdot g(\boldsymbol{x}_i, \boldsymbol{y}_i)}{\sum_{j=1}^{n} g(\boldsymbol{x}_i, \boldsymbol{y}_j)} \right] \tag{35}$$

Since we are drawing from the above distribution, $\boldsymbol{x}_i = \boldsymbol{y}_i$ is always true; therefore, we only need to enumerate how many $\boldsymbol{y}_j$ are different from $\boldsymbol{x}_i$ in order to compute one term of the expectation. In the case where we have $t$ pairs of $(1, 1)$ and $(n - t)$ pairs of $(0, 0)$, then for $g(1, 1) = g(0, 0) = 1$, $g(0, 1) = g(1, 0) = 0$ we have that:

$$\frac{1}{n} \sum_{i=1}^{n} \log \frac{n \cdot g(\boldsymbol{x}_i, \boldsymbol{y}_i)}{\sum_{j=1}^{n} g(\boldsymbol{x}_i, \boldsymbol{y}_j)} = \frac{1}{n} \left( t \log \frac{n}{t} + (n - t) \log \frac{n}{n - t} \right) \tag{36}$$

Moreover the probability of such an arrangement can be computed from the Binomial distribution

$$\Pr(t \text{ pairs of } (1, 1)) = \binom{n}{t} p^t (1 - p)^{n-t} \tag{37}$$

Therefore, we can compute the expectation that is $L(g)$ in closed form by computing the sum for $t$ from $0$ to $n$. We can apply a similar argument to computing the mean of ML-CPC values as well as the variance of the empirical estimates. This allows us to analytically compute the optimal value of the objective values, which allows us to perform direct comparisons over them.

## C.2 Mutual information estimation

The general procedure follows that in [40] and [44].

**Tasks** We sample each dimension of $(\boldsymbol{x}, \boldsymbol{y})$ independently from a correlated Gaussian with mean $0$ and correlation of $\rho$, where $\mathcal{X} = \mathcal{Y} = \mathbb{R}^{20}$. The true mutual information is computed as: $I(\boldsymbol{x}, \boldsymbol{y}) = -\frac{d}{2} \log \left( 1 - \frac{\rho}{2} \right)$ The initial mutual information is 2, and we increase the mutual information by 2 every $4k$ iterations.

**Architecture and training procedure** We consider two types of architectures – *joint* and *separable*. The *joint* architecture concatenates the inputs $\boldsymbol{x}, \boldsymbol{y}$, and then passes through a two layer MLP with 256 neurons in each layer with ReLU activations at each layer. The *separaable architecture* learns two separate neural networks for $\boldsymbol{x}$ and $\boldsymbol{y}$ (denoted as $g(\boldsymbol{x})$ and $h(\boldsymbol{y})$) and predicts $g(\boldsymbol{x})^\top h(\boldsymbol{y})$; $g$ and $h$ are two neural networks, each is a two layer MLP with 256 neurons in each layer with ReLU activations at each layer; the output of $g$ and $h$ are 32 dimensions. For all the cases, we use with the Adam optimizer [27] with learning rate $1 \times 10^{-3}$ and $\beta_1 = 0.9, \beta_2 = 0.999$ and train for $20k$ iterations with a batch size of 128.

## C.3 Knowledge distillation

The general procedure follows that in [47], where we use the same training hyperparameters. Specifically, we train for 240 epochs with the SGD optimizer with a momentum of $0.9$ and weight decay of $5 \times 10^{-4}$. We use a default initial learning rate of $0.1$, and divide the learning rate by 10 at 150, 180 and 210 epochs. We use $16384$ negative samples per positive sample [2], and a temperature of $0.07$ for the critic. We did not additiaonlly include the knowledge distillation loss to reduce potential compounding effects over the representation learning performance.

## C.4 Unsupervised representation learning

For CIFAR10, the general procedure follows that of MoCo-v2 [18, 10], with some slight changes adapted to CIFAR-10. First, we use the standard ResNet50 adaptation of $3 \times 3$ kernels instead of $7 \times 7$

kernels used for the larger resolution ImageNet, with representation learning dimension of 2048. Next, we use a temperature of $\tau = 0.07$, a batch size of 256 and a learning rate of 0.3 for the representation learner, and a learning rate of 3 for the linear classifier; we observe that these hyperparameters combinations is able to achieve higher performance on the CPC objective for CIFAR-10, so we use these for all our other experiments. The remaining hyperparameters are identical to the ImageNet setup for MoCo-v2. For ImageNet, we use the same procedure as that of MoCo-v2, except that we trained the representations for merely 30 epochs with a ResNet-18 network, instead of training on ResNet-50 for 800 epochs.

## Footnotes

[2]We note that this is smaller than what is used in [47], and it is possible to achieve additional (though not much) improvements by using more negative samples.