[Reviews · NeurIPS 2020]

Review 1

Summary and Contributions: The authors propose a multi-label version of contrastive predictive coding (CPC), which essentially transforms the CPC loss from one that is defined on each positive pair and its set of negative pairs for i=1...n versus a version where all positive pairs (and all possible negatives amongst them) make up the softmax distribution, which can be thought of a 'multi-label' classification task where the network is trained to ensure the top n predictions from that distribution are the n positive examples. The motivation behind this technique is that: - (1) The regular CPC loss (which is a lower bound on the mutual information between X and Y) is upper bounded by log(m), where m is the total number of pairs used (i.e. 1 positive pair + m-1 negative pairs). If log(m) is much lower than I(X;Y) then we underestimate mutual information. (In practice, m will be constrained by how much GPU memory / processing power is available.) - (2) alpha-CPC, the reweighted version of this loss (with weighting coefficient alpha) can upper bound I(X;Y) by log(m/alpha), potentially reducing bias, but also at the cost of increased variance. Its disadvantage however is that for some range of alpha the loss will no longer be a lower bound to I(X;Y). In essence, the authors show that for their proposed method (alpha-ML-CPC), one can derive the range of alphas that lower bound I(X;Y) as a function of m and n, and these ranges are quite large even for modest values of m and n. This means that all one has to do is find the right alpha within this range to balance bias and variance.

Strengths: - Technique is well justified, theoretically, with strong empirical results to support the theory. - The significance of the approach is that one can estimate tighter lower bounds to I(X;Y) without necessarily needing more computational resources (i.e. increasing m). This is significant because contrastive-based techniques generally require *many* negative pairs in order to achieve good performance.

Weaknesses: - While the paper certainly stands on its own merits even without the knowledge distillation results, I noticed that you are reporting test accuracy in both Table 1 and Table 2. It is not clear to me whether you (a) performed hyperparameter tuning on the validation set for each model enumeration (e.g. KD, CRD, L_1.0, L_0.1, J_1.0, etc.) and *then* got the best model for each and evaluated it on the test set; or (b) whether you just used the test set as your validation set. The latter approach (although seemingly common in papers nowadays) is not very well principled and can lead to optimistic results. It would be good if you can elaborate on this. - Table 1 and Table 2 should also report variances.

Correctness: I have not verified the proofs in the appendix, though the theoretical claims do appear to be well-supported empirically in the main text.

Clarity: Generally, yes. Some suggestions however: - The expectation symbol in your equations is redundant, since you already have the summation over n, so remove one and keep the other (I'd prefer keeping the summation). - In Equation (6), it would be much clearer if the summation outside the fraction was moved to the numerator. While either form is functionally equivalent, putting the summation in the numerator makes it more clear that your loss L(g) is one term computed over the entire batch, as opposed to Equation (2) where L(g) is defined on a per example-basis for i=1...n and averaged. Also, as mentioned earlier, this also erroneously has the expectation symbol. - Typo: "MC-CPC", should be "ML-CPC" - Typo: "we illustrates" should be "we illustrate"

Relation to Prior Work: Yes.

Reproducibility: Yes

Additional Feedback: - "two types of critic, named joint and separable" --> what is the difference between these two? - A very minor point related to the pdf itself: sometimes Figure 3 takes a while to render because of all the individual points in the plot (which is 20,000 per experiment, according to the x-axis). Perhaps these plots could be re-rendered by subsampling points or simply rasterising the plot to png (I would prefer the former, since this results in a much higher quality figure). ------ REVIEW UPDATE: I have read the other reviews and the rebuttal. I am satisfied with the response and will keep my score as-is.


Review 2

Summary and Contributions: Recent successful approaches to mutual information estimation and lower bounds use contrastive approaches, like contrastive predictive coding (CPC). This paper proposes an elegant modification of the CPC estimator that decreases bias with negligible computational cost. The paper demonstrates the value of the improved estimator for downstream tasks.

Strengths: There has been a lot of recent interest in the NeurIPS community in different information estimators. While CPC stands out as one of the more successful and widely used estimators, the fact that it saturates at log m, for m negative samples, is a persistent problem that highlights the bias in this estimator. This paper suggests a simple modification that improves bias and reduces the problem of saturating the bound with a small number of contrastive examples. The theoretical development is very clear, and includes conditions for the bound to hold. The empirical evaluation compares to CPC and shows both MI estimation as well as demonstrations of how it helps in downstream tasks. For the community of people interested in information bounds and estimators, this is certainly a welcome contribution.

Weaknesses: The scope of the paper is somewhat narrowly focused - only comparing directly with CPC at the expense of a menagerie of other variational estimators. However, I find it hard to fault this since CPC is the most widely used and is the natural comparison for this method.

Correctness: Yes.

Clarity: The paper was very clear and easy to follow.

Relation to Prior Work: This paper did a good job of summarizing recent work in this area.

Reproducibility: Yes

Additional Feedback: The plots are rather small and hard to read, maybe you can improve that in the final version with more space. Is the variance always strictly bigger? It's nice that you give an estimator with an adjustable range of bias-variance trade-off, I'm just wondering if there's a way to generalize even more so that Eq. 7 can somehow smoothly reduce to the standard CPC. Is it easy to understand why the variance goes up? I guess generically we expect that if bias is reduced it may increase variance, but the denominator in Eq. 7 adds together more R.V.'s, so I didn't expect the variance to be higher than CPC from that point of view. EDIT: I have read the author rebuttal.


Review 3

Summary and Contributions: This paper introduces a novel estimator based on a multi-label classification problem, where the critic needs to jointly identify multiple positive samples simultaneously. This is called multi-label contrastive predictive coding. With the same amount of negative samples, multi-label CPC can provably exceed the log(m)bound, while still being a valid lower bound of mutual information. Experiments in MI estimation, knowledge distillation, self-supervised learning demonstrate the validity of proposed methods. -------------------------------------- Updates: The rebuttal addresses my concerns and thus I recommend to accept this paper.

Strengths: + A well-written paper + Extensive experiments + Nice theoretical justification + The idea of generalized CPC and ML-CPC is interesting.

Weaknesses: - The experimental improvement comparing to CPC is marginal. However, since the ML-CPC has similar computation complexity, ML-CPC is still valuable. - Lacking results on ImageNet. I think the results on ImagetNet is very important. For most knowledge distillation and self-supervised learning works, ImageNet is a standard benchmark. Thus to make this work more convincing, I suggest the authors to report results on ImageNet, especially considering the marginal improvement on CIFAR. - Typo: Line 133: xxx negative pairs "share" a same element xxx - The multi-label CPC is provably lower than MI. However, in Fig. 3, the curves of ML-CPC (\alpha=0.001, 0.0001) without smoothing surpasses the MI. Can you provide some explanation?

Correctness: The method and claims are correct.

Clarity: Yes

Relation to Prior Work: Yes

Reproducibility: Yes

Additional Feedback: If the results of ImageNet are presented, I am happy to raise scores.


Review 4

Summary and Contributions: Contrastive Predictive Coding (CPC) estimates are upper bounded by log of negative samples per positive samples, while computational overhead limits the increasement of the number of negative samples. Aim at this problem, this paper proposes ML-CPC for increasing the bound without additional computational costs and decreasing bias. **The authors have carefully addressed my questions in the rebuttal. It's very rewarding to reading this paper.**

Strengths: 1. This paper explores a practical problem and solves it by proposing three generalizations to CPC (re-weighted CPC, ML-CPC and re-weighted ML-CPC). These methods are easy and step-by-step. 2. This paper proves the learning objective of ML-CPC is still guaranteed to be a variational lower bound to mutual information and has similar computational costs to CPC. 3. The paper provides a thorough experimental validation of the proposed methods, and the results align with their claims.

Weaknesses: I have a few minor concerns: 1. In Sec. 2, the authors say "Therefore, one can train g and h to maximize L(g), resulting in higher lower bounds to I(X;Y)...", should it be "minimize L(g)"? 2. In Sec. 3.1 and Sec. 3.3, I don't very understand why L_{\algph}(g) is upper bounded by log(m/\alpha). It may need an explanation.

Correctness: The problem is well-motivated and the technical content seems to be sound.

Clarity: The paper is clearly written and fairly readable.

Relation to Prior Work: The authors review the previous works and naturally raise their motivation. The main contribution of this paper is to propose a method decreasing bias and being a lower bound to mutual information. In my knowledge, this is a novel work.

Reproducibility: Yes

Additional Feedback:

[Author Response · NeurIPS 2020]

We thank the reviewers for their constructive feedback. All the reviewers agree that the proposed method, multi-label contrastive predictive coding (ML-CPC), leads to mutual information estimators with lower bias at negligible computational cost, and its value is demonstrated empirically on mutual information estimation, knowledge distillation and unsupervised representation learning. Here, we answer some major questions raised by the reviewers.

**[R1] Did you tune for each case separately in the knowledge distillation experiments?** No. We use the default hyperparameter setting of $a = 0.0$ (which is without KD loss) and $b = 0.8$ (which is the weight of distillation loss against classification loss) for all CPC and ML-CPC experiments. This is the default setting given by the CRD implementation here[1], and we did not tune that specifically for our case. We will clarify this in the final version.

**[R1] Please include variances for KD results.** Thanks for the suggestion. Our results are averaged over 3 seeds, but unfortunately we were not able to include variances due to width limitations, which is around $0.2$ for each case. We will include variances in the supplementary material and add a reference to this in the main paper in the final version.

**[R1] The expectation symbol seems redundant.** We believe this is necessary since our lower bound arguments would rely on taking the expectation; the objective value for some mini-batch is not necessarily a lower bound to the mutual information of the entire distribution.

**[R1] Joint and separable critic?** Our setup follows that of [Poole et al. 2019][2]. "Joint" means a single neural network $f(x, y)$ is used, while "separable" means that the inner product of two separate neural networks $g(x)^\top h(y)$ is used. "Joint" is more flexible, but requires $O(NM)$ network evaluations for each batch, while "separable" is less flexible but only requires $O(N + M)$ evaluations. We will explain this more clearly in the final version.

**[R2] Why is ML-CPC expected to have higher variance than CPC?** It is indeed difficult to provably compare the variance *between ML-CPC and CPC* since under finite $n$ and $m$, it is hard to obtain the optimal critic in closed form, except for some special cases (although for the same $\alpha$, the variances seem similar). However, we believe that for $\alpha_1 < \alpha_2$, the variance of $\alpha_1$-ML-CPC tends to be larger than that of $\alpha_2$-ML-CPC (and similar for CPC). Apart from the usual bias-variance trade-off argument that you mentioned, our intuition for this is as follows. The target of the (ML)-CPC objectives is to estimate density ratio $r(x, y) = p(x, y)/p(x)p(y)$ with

$$\frac{g(x, y)}{\frac{\alpha}{m} g(x, y) + \sum g(x, \overline{y})}$$

where smaller $\alpha$ indicates that the estimates are more flexible (i.e. being able to take larger values), so this estimate becomes larger for large $r(x, y)$ and relatively the same for small $r(x, y)$, which increases variance. We will add a discussion about this in the final version.

**[R3] ImageNet results.** We tried modifying the knowledge distillation code for ImageNet (since the code for ImageNet is not provided out-of-the-box), but were unable to fully reproduce the results in the CRD paper, so the results here remain inconclusive. As of now, we are still running the representation learning experiments, which takes days with 4 GPUs; we will update ImageNet results in the final version, where we would have more time to run the experiments.

**[R3] Why do ML-CPC estimates surpass MI without smoothing for $\alpha = 0.001, 0.0001$?** First, while our results guarantee that the expectation of batch-based estimates to be a lower bound, and we optimize an unbiased estimate to the lower bound via SGD, it is still possible for estimates of particular batches to exceed MI. Second, our theory guarantees the lower bound property for $\alpha \approx 1/128$ (since batch size is 128), so for $\alpha = 0.001, 0.0001$ we do not have guarantees. Nevertheless, we wished to demonstrate that these choices of $\alpha$ can still be used empirically; we will clarify this in the final version.

**[R4] "In Sec 2, ... should it be "minimize $L(g)$"?**

It is indeed a maximization. Our goal is to "maximize" mutual information through unbiased estimates of a lower bound to it; the setup is analogous to maximizing the "evidence lower bound" in variational inference.

**[R4] "... why $L_\alpha(g)$ is upper bounded by $\log(m/\alpha)$"?** Because our critic $g$ outputs only positive values, the division in the expectation satisfies:

$$\frac{g(x, y)}{\frac{\alpha}{m} g(x, y) + \sum g(x, \overline{y})} \leq \frac{g(x, y)}{\frac{\alpha}{m} g(x, y)} = \frac{\alpha}{m}$$

Taking the expectation over values that are not greater than $\log(m/\alpha)$ then proves the claim. We will add a lemma to support this claim in the final version.

**[R1] [R2] [R3] Typos and Figures** Thank you for the suggestion; we will fix these issues in the final version.

## Footnotes

[1] https://github.com/HobbitLong/RepDistiller/blob/master/scripts/run_cifar_distill.sh#L29

[2] On Variational Lower Bounds of Mutual Information, ICML 2019


[Meta-Review · NeurIPS 2020]

This paper was highly regarded by all reviewers. I looked at the paper myself because of a possible conflict between the claimed lower bound results and formal limitations on the measurement of mutual information proved in [McAllester and Stratos, AISTATS, 2020] which the authors cite. The paper emphasizes that ML-CPC improves the upper bound on MI estimator from log(n) in traditional CPC to (roughly) log(n^2) (for modest m and alpha at the lower end of its range). This is in rough correspondence with the the completely general upper bounds on MI estimation of McAllester and Stratos. However, a failure to prove a tighter upper bound on ML-CPC does not imply that no tighter upper bound exists. A more convincing, and perhaps enlightening result would be to demonstrate a case where ML-CPC achieves this upper bound --- that a lower bound on MI of size log n^2 is actually achievable. But the empirical results are strong in any case.